# Polarization Induced Electro-Functionalization of Pore Walls: A Contactless Technology

**DOI:** 10.3390/bios9040121

**Published:** 2019-10-11

**Authors:** Aurélie Bouchet-Spinelli, Emeline Descamps, Jie Liu, Abdulghani Ismail, Pascale Pham, François Chatelain, Thierry Leïchlé, Loïc Leroy, Patrice Noël Marche, Camille Raillon, André Roget, Yoann Roupioz, Neso Sojic, Arnaud Buhot, Vincent Haguet, Thierry Livache, Pascal Mailley

**Affiliations:** 1Univ. Grenoble Alpes, CEA, CNRS, IRIG, SyMMES, F-38000 Grenoble, France; liujie1130@gmail.com (J.L.); loic.leroy@univ-grenoble-alpes.fr (L.L.); camille.raillon@cea.fr (C.R.); andreroget2009@hotmail.fr (A.R.); yoann.roupioz@cea.fr (Y.R.); arnaud.buhot@cea.fr (A.B.); Thierry.livache@aryballe.com (T.L.); 2LAAS-CNRS, Université de Toulouse, 31400 Toulouse, France; emeline.descamps@inserm.fr (E.D.); tleichle@laas.fr (T.L.); 3Department of Chemistry, University of Oxford, South Parks Road, Oxford OX1 3QZ, UK; ismail.abdulghani@gmail.com; 4Univ. Grenoble Alpes, CEA, LETI, MINATEC Campus, 38000 Grenoble, France; pascale.pham@cea.fr (P.P.);; 5CEA/DRF/IRIG, Centre de Thérapie Cellulaire, Hôpital Saint-Louis, F-75010 Paris, France; Francois.chatelain@cea.fr; 6Institute for Advanced Biosciences, Grenoble Alpes University/INSERM U1209/CNRS UMR5309, 38700 La Tronche, France; patrice.marche@univ-grenoble-alpes.fr; 7University Bordeaux, CNRS, Bordeaux INP, ISM, UMR 5255, F-33400 Talence, France; neso.sojic@enscbp.fr; 8Univ. Grenoble Alpes, CEA, INSERM, IRIG, BGE, F-38000 Grenoble, France; vincent.haguet@cea.fr

**Keywords:** nanopore, micropore, CLEF, biosensing, electro-functionalization, contactless

## Abstract

This review summarizes recent advances in micro- and nanopore technologies with a focus on the functionalization of pores using a promising method named contactless electro-functionalization (CLEF). CLEF enables the localized grafting of electroactive entities onto the inner wall of a micro- or nano-sized pore in a solid-state silicon/silicon oxide membrane. A voltage or electrical current applied across the pore induces the surface functionalization by electroactive entities exclusively on the inside pore wall, which is a significant improvement over existing methods. CLEF’s mechanism is based on the polarization of a sandwich-like silicon/silicon oxide membrane, creating electronic pathways between the core silicon and the electrolyte. Correlation between numerical simulations and experiments have validated this hypothesis. CLEF-induced micro- and nanopores functionalized with antibodies or oligonucleotides were successfully used for the detection and identification of cells and are promising sensitive biosensors. This technology could soon be successfully applied to planar configurations of pores, such as restrictions in microfluidic channels.

## 1. Introduction

Over the last twenty years, functionalized tridimensional pores have emerged as a specific range of biosensors offering sensitivities higher than those of conventional methods [1,2,3,4]. Pore sensing was successfully employed to detect and analyze different types of biomolecules and cells: single- and double-stranded nucleic acids [5], peptides [6], proteins [7], bacteria [8,9], viruses [10,11], and cancer cells [12,13]. Scheme 1 describes the principle of detection of a single biomolecule passing through a pore: When crossing the pore, the target partially blocks the aperture (Scheme 1A), which is detected by a variation in the ionic current (Scheme 1B). The variations of the transit time (Δt) and electric current intensity (ΔI) provide information about the length and diameter of the target, respectively, which can be used to distinguish it from other molecules present in the assay medium. To achieve high sensing sensitivity, the pore diameter is preferably chosen to be slightly greater than that of the target to optimize signal-to-noise ratio. The advantages of the pore sensing technique are: (i) the detection of single (bio)molecules, (ii) without labeling, and (iii) via a quick and inexpensive electrical methodology.

Two main types of pores are considered: biological and solid-state pores. This review describes the principle of pore sensing, with a focus on solid-state pores, and how our contactless electro-functionalization (CLEF) technology can overcome the limitations of the current functionalization strategies.

## 2. Principle of Pore Sensing and Problematics of Functionalization

### 2.1. Biological Pores

Biological nanopores are mainly formed from proteins, peptides, or DNAs. Initial experiments using the principle of electrical biosensing through pores were performed with a bacterial protein nanopore, the α-hemolysin, integrated in a planar lipid bilayer, thereby reproducing a natural ionic channel [14]. The efficiency and robustness of detection of polynucleotides using α-hemolysin nanopores were confirmed by several teams [14,15,16,17]. To improve the sensing resolution of the pores, a passage from the cylindrical α-hemolysin to the sharper *Mycobacterium smegmatis* (MspA) [18,19] and *Escherichia coli* CsgG proteins was envisioned [20,21]. Recently, down to two-base long oligonucleotides have been resolved using an aerolysin protein nanopore [22]. In addition, thanks to molecular biology techniques, specific receptors were introduced at various sites within the protein nanopore by molecular biology techniques in order to promote a specific interaction with the target [23,24]. These modifications extend the electrical detection capability of protein nanopores to other targets, such as heavy or even small organic molecules or metal ions [25]. All the advantages of protein nanopores, which range from defined and stable scaffolds to the possibility of targeted amino acid modifications and simple engineering to module the inherent characteristics [26], have led to their commercialization. In 2012, Oxford Nanopore Technologies introduced the first nanopore-based sequencer, MinION^®^, a device holding 500 protein nanopores [27,28,29]. The main advantage of the MinION technology is that it allows very long reads (>150 kbp) [30].

However, electrical biosensing using protein nanopores presents some limitations. The protein is included in a lipid bilayer isolating the two sides of the pore. The lipid bilayer is neither mechanically nor electrically stable [31]. Several approaches have been conducted to overcome this inherent limitation such as the inclusion of polymerizable lipids [32,33], the use of hydrogels and inorganic supports [34,35], reduction of the lateral bilayer size [36], ‘droplet interface bilayers’ (DIBs) [37,38], and replacement of the lipids by amphiphilic polymers [26]. The protein itself is not very stable and has a relatively short lifetime for detection as a result of the sensitivity of the protein to temperature, voltage, ion concentrations, and solvents [39,40]. These nanopores cannot therefore be used for detection over long periods of time. Moreover, the diameter and geometry of the available protein nanopores are in the order of a few nanometers (few are more than 5 nm), limiting their scope of sensing to unfolded proteins or single-stranded DNA [41]. Although targeted amino acid modification is possible, it is still limited to a small number of amino acids and large parts of protein could not be simply deleted or de novo fabricated using non-natural amino acids [26]. Careful manipulation is also required to form the lipid bilayer and to integrate the protein nanopores in the desired location. Coupled with the instability of the bilayer, the integration of the protein nanopore into a microfluidic system is challenging.

To overcome the limitations of protein nanopores, especially to more simply achieve modulation of the pore geometry and attachment of chemical functions at their core, nanopores based on peptides [42,43] and DNA origami were developed [44,45,46]. Polypeptide nanopores are very limited in terms of the dimensions of the lumen of the nanopore (<1.5 nm) and in terms of the number of amino acids (50). The importance of DNA origami in designing nanopores over the polypeptide nanopores is mainly in the possibility of modulating the nanopore diameter to more than 20 nm. However, the possible repertoire of DNA is limited to four DNA bases. DNA nanopores with atomically defined structures of predictable nanomechanical properties have been used for sensing and for controlled drug release thanks to the possibility of their gating [47,48,49]. In order to be incorporated in the lipid bilayer, negative DNA origami should be engineered in order to carry a lipidic molecule capable of integrating it into the membrane [50]. An alternative method by engineering of non-negative DNA is applied to avoid lipid anchoring [51]. The limitation of these DNA nanopores comes from their complex anchoring to the biological lipid membrane with its inherent increased leakage and structural fluctuation of DNA nanopores compared to protein nanopores [52,53].

### 2.2. Solid-State Pores

Thanks to advances in lithography and etching, synthetic nanopores with controlled diameters have been successfully fabricated in solid-state membranes [1,3,4]. Solid-state nanopores, similar to their biological counterparts, are nanometer-sized apertures, made in thin synthetic films or thicknesses ranging from a few nanometers to several micrometers. Synthetic nanopores are a promising alternative because a pore in a solid-state membrane overcomes almost all the drawbacks of biological nanopores [10,39]: (i) The pores are mechanically stable over time, even in the presence of electric fields; (ii) they are insensitive to variations of temperature, pH, and salt concentrations; (iii) the pore diameter can be precisely controlled, with an accuracy in the order of 1 nm for the nanopores, (iv) the number of pores per unit area can be precisely controlled, which is of great importance for single-molecule detection [54], and (v) larger surface areas and mass production capabilities, which are a requisite for commercialization, could be made easier than with biological pores [55].

Depending on the type of membrane material and the pore diameter, the manufacturing technique can be either chemical etching [56], ion-track etching [57], ion beam sculpting [58], helium ion beam [59], controlled dielectric breakdown [60], laser ablation [61], controlled optical etching [62], or electron beam carving [63]. The membranes can be made of polymers, such as polyethylene terephthalate (PET) [64] or polycarbonate [65], but they are most often made from inorganic materials such as silicon nitride (Si_3_N_4_) [66], silicon oxide (SiO_2_) [63], alumina (Al_2_O_3_) [67], or hafnium oxide (HfO_2_) [68]. Despite the very robust detection obtained with nanopores fabricated in these materials, their key problem was the high thickness of the membrane which can limit the resolution of sensing [69]. However, crystalline atomically thin 2D materials have been developed and integrated as membranes for nanopores. Among others, we may mention 2D materials such as graphene [70,71], molybdenum disulfide (MoS_2_) [72], heterostructure of graphene and MoS_2_ [73], boron nitride (BN) [74], and tungsten disulfide (WS_2_) [75]. Despite their advantages of very low thicknesses, these 2D materials suffered from several drawbacks that prevented their implementation for DNA sequencing, among which were noise upon sensing and mechanical fluctuations [54,69]. More recently, Mojtabavi et al. investigated the use of atomically thin flakes of 2D transition metal carbides called MXenes as supports for nanopore sensing [76]. Solid-state nanopores were used in bioassays to discriminate DNA fragments of different lengths [64] and for distinguishing single-stranded from double-stranded DNAs [77]. Moreover, larger-diameter nanopores could distinguish proteins of different sizes [78,79] or investigate the processes of particle translocation [80,81]. 

Biological analysis with functionalized synthetic nanopores presents significant advantages over non-functionalized nanopores. Pore functionalization can provide a variety of changes in the physical and chemical properties (i.e., selectivity, hydrophobicity, surface charges, and specific molecular recognition), and, thus, the ionic transport properties are modified. The fixation of a specific probe within the pore makes the distinction of targets of comparable size become possible: When immobilizing single-stranded DNA probes in a nanopore, the complementary DNA strand can be discriminated from other targets with single mismatch specificity [82]. Furthermore, a nanopore functionalized with an antibody can selectively detect a target protein [83]. Biomimetic ionic nano-channels have also been made in a PET membrane by functionalizing the nanopore with DNA single strands capable of selectively generating quadruplexed DNA nanostructures in the presence of potassium ions [84].

### 2.3. Functionalization of Synthetic Pores and Limitations of Current Techniques

Biological nanopores exhibit high selectivity due to the presence of functional groups at their inner walls, which permits the traversing analytes to interact with the pore wall. Consequently, accelerating, slowing, or preventing the passage of analytes by biomolecular interactions inside the pore make it possible to differentiate between analytes [85]. For instance, some bacterial membrane protein nanopores are very selective to some kinds of oligosaccharides [86] or do not permit passage except for one selective analyte [87]. Similarly to what was done for protein nanopores, the local functionalization of the inner wall of synthetic nanopores is necessary to expand their biomolecule detection capability. The conventional methods for functionalization of “smart nanopores” rely on one of the following strategies: (i) deposition techniques (chemical and physical vapor deposition [88,89,90], electroless deposition [91], and atomic layer deposition [92]), (ii) chemisorption of functional molecules using thiol–gold or silanes [93], (iii) chemical modification of the functional group on the nanopore to yield polymer brushes [94] or hydrogels [95], and (iv) plasma surface modification [84]. The functionalization of the inner walls of tridimensional nanopores is nevertheless a major technological challenge and a current barrier to their use. The previously listed conventional methods of surface chemistry, adapted to the nature of the constituent material, are however imperfectly capable of tackling this task. Indeed, in addition to pore functionalization, the surrounding membrane is also functionalized [96], which significantly reduces the sensitivity of the biosensing technique: When the molecular probes are also immobilized on the membrane surface, a very large part of the species to be detected is captured and sequestered on the membrane, which leads to a significant loss of sensitivity regarding pore sensing. Localized functionalization of the inner face of the nanopore in two steps, comprising first the activation of the inner surface of the nanopore, followed by grafting, has been described [97]. To do so, a silica layer was first deposited in a nanopore in a silicon nitride membrane, using an electron beam in an environment containing steam tetraethyl orthosilicate. This method introduced a silanol termination on the silica surface of pore walls, thus localizing the following silanization step on the walls of the nanopore. In addition to the difficulty of implementing this technique, this approach offers very little geometric accuracy during silica deposition. This method of localized functionalization of nanopore walls is therefore expensive and difficult to implement.

### 2.4. The CLEF Technology

#### 2.4.1. Principle of Contactless Electro-Functionalization (CLEF)

In 2009, we proposed an innovative functionalization technique of tridimensional pores, inspired by bipolar electrochemistry methodologies. This process, called “contactless electro-functionalization” (CLEF) [98], allows the selective functionalization of the inner wall of micro- and nanopores manufactured in a solid-state semiconductor membrane (Figure 1).

Briefly, as schematized in Figure 1A, the application of an electric field through a unique pore etched in a silicon membrane (Figure 1B) leads to the functionalization of its inner silica walls with electroactive species present in the surrounding media. No deposition is observed in the absence of a voltage drop applied between the two electrodes present on each side of the pore. This methodology has proved to be rather versatile since it enabled the functionalization of pore walls with electrodeposited films or objects independently of the involved redox phenomena (i.e., anodic or cathodic electrodeposition), leading to the deposition of iridium oxide (Figure 1C), gold nanoparticles (Figure 1D), and polypyrrole-bearing oligodesoxyribonucleotide (PPy-ODN) (Figure 1E,F). To illustrate the robustness of pore functionalization, we submitted PPy-ODN deposits to recognition/revelation/washing cycles using complementary DNA strands [98]. Up to four cycles were carried out without significant loss of fluorescence response despite the harsh denaturation conditions, thus demonstrating the robustness of the electrodeposited polymer film. 

Moreover, the CLEF deposition process can be applied to a wide range of pore diameters (from 70 µm down to 50 nm), and, as displayed in Figure 1D,E, appears to be independent of the surface roughness of the pore walls. As seen in Figure 1E, and as expected with bipolar processes, the electro-induced deposition process is dependent on the field vector, as the micropore exit (according to the picture orientation) is selectively functionalized with the polypyrrole layer. 

Two hypotheses were initially proposed for the process of surface functionalization [98]. First, the micropore inner walls could exhibit their own conductivity and behave like a dipole, creating a bipolar effect through the formation of a couple of electrodes at each side of the pore. The second hypothesis relied on the role of oxygen that could be activated under these specific conditions. However, both hypotheses were invalidated by the existence of radial growth of iridium oxide films in oxygen-free CLEF experiments. Nevertheless, Figure 2 depicting the distributions of the electric field in two configurations of solid-state micropores clearly suggests the existence of electrochemical reactions at the silica/electrolyte interface. Therefore, numerical modeling of the overall system was carried out to better identify important features associated with the CLEF configuration such as electrical field localization and distribution within the solution and the silicon substrate or influence of the membrane composition and its geometric parameters.

#### 2.4.2. CLEF’s Mechanism

##### Numerical Simulations of the Electric Field Distribution Inside a Pore

When a voltage is applied across a micropore, a high electric field intensity is expected inside the pore due to the geometric restriction (Figure 2). In order to investigate the electric field distribution in the system, numerical simulations were performed using a 2D axisymmetric finite element model with COMSOL Multiphysics™ [99]. The geometry of the micropore is illustrated in Figure 2A: A micropore of 15 µm diameter was manufactured in a silicon membrane covered by a silica layer of 4 µm thickness. The mathematical underlying model was based on the complex electrokinetic equation which takes into account both the conductive and capacitive common properties of the materials (electrolyte, pore membrane composed of silicon and silicon dioxide [100]). The electrolyte conductivities used in the numerical model were measured. The Ag/AgCl electrodes were considered as perfect electrodes so that the applied potential was defined at their boundaries.

This numerical model revealed the influence of the membrane on the electrical field distribution inside the pore: If the pore membrane is considered as a perfect insulator, numerical simulations confirm that the electric field is tangential to the pore membrane (Figure 2B right). However, CLEF does not work under this experimental configuration. It means that a more complex membrane structure is necessary to provoke the CLEF phenomenon: The “sandwich-like” structure of the Si/SiO_2_ membrane proves to be a key parameter in allowing electrochemical grafting onto the pore walls [99]. When the electrical properties of the inner core silicon layer are included in the numerical model, numerical simulations show that the electrical field has a non-zero component normal to the pore membrane (Figure 2C right).

These simulation results suggest that an electrochemical grafting at the surface of the inner walls of a pore is possible only if an electrical pathway is created between the electrolyte and the membrane silicon core through the SiO_2_ layer, creating in this way a possible exchange of electrons between the electrolyte and the conducting core silicon. This pathway could be caused by defaults in the silicon oxide layer or by high electrical voltage.

##### Validation of the Numerical Model Using Impedance Measurements

Impedance measurements (Bode plots) were performed using a commercial impedance spectrometer (Biologic SP300) in the frequency range 1–7 MHz, by applying a potential of 100 mV to the two Ag/AgCl electrodes located at each side of the pore [99]. The resistivity of the inner silicon core was 0.010−0.025 Ω·cm. Figure 3 compares the measured impedance (Bode plots) with the numerical impedance computed from the previous numerical simulations (Figure 2C).

The numerical impedance norm showed good agreement with the experimental measurements as supported by the similarity observed between the curve shapes, validating the developed numerical model (Figure 3A). As the electrical capacity is a function of the square root of the electrolyte concentration in the diffuse layer, the frequency increases logically with growing KCl concentrations. However, a frequency shift between the theoretical curve and experimental data was observed at all KCl concentrations. This must be due to the fact that the simulation conditions are ideal and may not take into account environmental or material factors (pH, quality of silica, etc.). 

However, a capacitive effect (impedance phase around −90°) decreased the impedance norm above a cutoff frequency whose value does not coincide with the numerical model (Figure 3B). This capacitive effect may come from the Si/SiO_2_ interface, from the SiO_2_/electrolyte interface or from the silicon light illumination. Despite this difference in the cutoff frequency, this numerical model can provide indicative explanations of the role of each material in the CLEF mechanism.

#### 2.4.3. Importance of the Core Silicon

Our results suggested a predominant role of the silicon core of the membrane in the bipolar process (Figure 2C), and further exploration of its influence in the CLEF process needed to be undertaken. However, for manufacturing reasons, experiments could only be carried out with one membrane thickness. We therefore used our numerical model to simulate the effect of other thicknesses [99], and the obtained data are reported in Figure 4A.

The phase component of the membrane impedance strongly differs for solid-state membrane only constituted of silica and for Si/SiO_2_ membranes (Figure 4A). No difference is seen for silicon cores exhibiting thicknesses ranging from 1 to 2 µm. Importantly, some electro-functionalization experiments were run on purely insulating membranes (silicon nitride membranes) and led to no surface functionalization (data not shown). Such contrasting behaviors clearly highlight the major role of the semiconductive silicon core on the electric field repartition. 

Since silicon is a material for which both doping and lighting may induce significant changes in its semiconductive properties, these two conditions may also have an influence on the CLEF process. The influence of light on the CLEF process was especially assayed through the application of the functionalization process using the same current density either in the dark or under white light illumination (data not shown). The putative electro-induced deposition of PPy-ODN was revealed by hybridization with a fluorescently labeled complementary DNA strand. For the electro-functionalization process performed in the dark, no fluorescence was detected, whereas under illumination a strong ring-shaped fluorescence signal was observed with a diameter corresponding to the pore size. This observation highlights again the role of the silicon core. Presumably, the illumination could provoke an increase in the charge carrier density in the membrane through the generation of electron–hole pairs that are firmly separated by the electric field, thus increasing the conductivity of the membrane silicon core. 

Under a certain resistance threshold, the silicon membrane may not contribute actively in the charge transport within the system. In other words, the CLEF process only appears if the electric field lines are not confined within the electrolytic pathway present in the pore. The amplitude of the applied electric field was therefore varied to investigate its influence on PPy-ODN bipolar deposition. Three different current pulse intensities (2, 5, and 11 µA) were applied across 18 µm-wide micropores. No deposition was obtained for the smaller current whereas fluorescence was observed for the higher ones (Figure 4B), confirming the existence of an electric field amplitude threshold above which contactless electro-functionalization on the pore walls is obtained.

In addition, increased electric field amplitudes generated higher fluorescence levels, suggesting larger deposited quantities. At higher current amplitudes, the fluorescence image showed local spreading onto the membrane at the circumference of the micropore (Figure 4B). To confirm that this larger fluorescence ring was not simply due to fluorescence diffusion linked to a higher fluorescence signal, polypyrrole-amine was deposited on the pore walls using CLEF. The primary amines on the pore walls were then used for gold nanoparticle fishing so that the trapped gold nanobeads could be used as localization revelators of the CLEF deposits using SEM (Figure 4C–E). No bead deposition was observed for pulses of 2 µA, confirming the existence of an electric field amplitude threshold (Figure 4C). Furthermore, the deposit is partly localized on the membrane for electro-functionalization at higher electric field amplitudes (Figure 4D,E). The depositions of gold nanobeads confirm those obtained by fluorescence and show that, for a similar pulse duration, depositing larger quantities by applying higher currents comes at the cost of a weaker localization of the grafting, which is no longer localized to the pore walls.

#### 2.4.4. Geometric Effects of the Pore Walls

In CLEF, the applied electric field is directional during the deposition process, and this polarization is illustrated in the model presented in Figure 2C. In this model, we have integrated the scalloped inner wall profile that is generated during the etching process (deep reactive ion etching). This wall roughness creates “needle” effects on the surface with a few reversals of the wall polarization along the pore depth. Polarization reversals on pore roughness may explain the full coverage of the pore walls.

To investigate this effect, CLEF was applied onto a pore exhibiting a smooth surface with a ridge in the middle [101]. In contrast to the scalloped pore wall (Figure 2C on the right side), no polarization effects were seen on the surface (Figure 5A). Consequently, a strong isotropy between pore entry and exit appeared to exist. Such a configuration leads to non-symmetric pore wall functionalization with electrodeposited gold particles on the cathodic side of the pore (Figure 5B,C). Again, a strong correlation between the model prediction and experimental pore wall decoration was obtained, highlighting the existence of the radial polarization of the pore entries.

CLEF can also be operated either under potentiostatic or galvanostatic control [101]. Depending on the applied stimulation, metal electrodeposition is obtained either as rods crossing the diameter at the cathodic side of the pore under galvanostatic control (Figure 6A) or as nanoparticles located along the overall surface for the potentiostatic control (Figure 6B). The effective potential across the pore is variable under potentiostatic control, thus, favoring nucleation behavior. On the opposite, by maintaining a constant current, the potential across the pore is maintained stable, leading to a growing process instead of nucleation. The latter experiment highlights how CLEF can also control the grafting morphology by varying the stimulation protocol.

### 2.5. Applications in Detection

The possibility of using CLEF-functionalized micro- and nanopores for biosensing was demonstrated using PPy-ODN probes grafted on the inside walls of solid-state nano- and micropores.

#### 2.5.1. Detection of Bio-Functionalized Particles in Nano- and Micropores

Solid-state nanopores were functionalized with ODNs using CLEF [97]. Nanopores of 200 nm in diameter were covered with PPy-ODN and then saturated with BSA in order to minimize non-specific adsorption (Figure 7A). Gold nanoparticles (AuNPs, 100 nm in diameter) bearing complementary or non-complementary ODN sequences (c-ODN-AuNPs and nc-ODN-AuNPs, respectively) were transported through the pore using a stationary electric field. SEM pictures of functionalized nanopores revealed a higher density of cODN-AuNPs close to the ODN-functionalized pore compared to the naked pore, suggesting hybridization between pore and particle ODNs (Figure 7B). 

Figure 7C shows the scatter plots of current amplitude variations, ΔI, versus translocation time, Δt, of ODN-coated gold nanoparticles. The ΔI distributions were similar for c-ODN-AuNPs and nc-ODN-AuNPs, confirming that these two populations of nanoparticles had a similar diameter. However, the Δt distribution was dramatically different: c-ODN-AuNPs showed higher Δt values than nc-AuNPs, reflecting the fact that c-ODN-AuNPs were slowed down or temporarily stopped in the pore because of biochemical recognitions, contrary to nc-ODN-AuNPs which passed through much faster and did not hybridize with grafted ODNs. The average translocation time for c-ODN-AuNPs was 490 ms whereas it was only 2.2 ms for nc-ODN-AuNPs. The Δt values showed considerable variability, no doubt due to a huge variation in the number of specific interactions established between c-ODN-AuNPs and the pore walls during transit and due to not perfectly identical nanoparticle velocities within the pore. These results show the potential of the CLEF technology in providing selective and specific nanopore biosensors and extend the application of CLEF-functionalized nanopores to the detection of nanometric objects such as nanoparticles and biomolecules.

Investigations about the transit of 20 µm-wide polystyrene microparticles through micropores were also carried out as a proof of concept for the biosensing capabilities of CLEF-modified micropores for cell detection [101]. PPy-ODN-modified micropores were incubated with complementary ODN-modified polystyrene (c-ODN-PS) particles. Observations by optical transmission microscopy and monitoring of the variation of the ionic current, in real time, confirmed that c-ODN-PS particles were immobilized in ODN-functionalized micropores whereas no capture of non-complementary ODN-modified polystyrene (nc-ODN-PS) particles was observed. Figure 8A shows short drops in current intensity corresponding to the transit of nc-ODN-PS particles through the pore with no interaction with ODNs on its walls. On the opposite, the long drop in current intensity in Figure 8B suggested pore blockage by a c-ODN-PS particle interacting with grafted ODNs. 

#### 2.5.2. Cell Capture and Identification in Antibody-Functionalized Micropores

In a more complex approach, a micropore biosensor was conceived to study the passage of living cells [102]. ODN-modified 15 µm-large micropores were converted into antibody-modified micropores by using antibody–ODN conjugates, as previously described [39,103,104,105] (Figure 9A). Mouse spleen cells, containing a mixture of B- and T-lymphocytes, were used as a biological model to evaluate the recognition properties of the functionalized micropores. B- and T-lymphocytes are undistinguishable in optical microscopy without prior labeling because of their similar morphologies. However, they express different markers at their surface that can be recognized by specific antibodies, i.e., anti-CD19 and anti-CD90 for B- and T-cells, respectively. In the experiment, for visualization purposes, only T-lymphocytes were selectively labeled with R-phycoerythrin conjugated with an anti-CD3 antibody. The devices were positioned over an inverted fluorescence microscope allowing visual access to the micropore (Figure 9B). Two behaviors could be observed in functionalized micropores: The cells were either translocated through the pore or remained trapped inside the pore. Epifluorescence microscopy revealed that T-lymphocytes were trapped in anti-CD90-modified micropores and B-cells in anti-CD19-functionalized micropores, respectively. No cell capture could be observed in non-functionalized or ODN-modified control micropores. The CLEF-functionalized micropores can therefore be useful tools for ensuring selective capture of well-defined cell types. The challenging capture and identification from samples composed of complex mixtures of biological objects with similar sizes and morphologies without labeling have been successfully achieved using CLEF-functionalized pores.

### 2.6. From “Through” to “Planar” Pores

As discussed in Section 2.2, micro- and nanopores are fabricated by drilling an aperture in a solid-state membrane. This kind of “through” pore suffers from complex handling in terms of fluidic connections coupled with electrical connections. Recently, Long and collaborators reported the advantage of combining bipolar electrochemistry with pore technology [22,106,107,108,109] through the development of a metal-coated wireless nanopore electrode for the detection of single small molecules and ions [107] and the real-time monitoring of NADH in living cells [108]. It is therefore relevant to shift from “through” to “planar” pores, i.e., restrictions inside microfluidic channels. These restrictions are easy to fabricate using substrates such as silicon, silicon oxide, and PDMS and can have similar applications as through micropores [110,111,112,113,114,115]. Such restrictions permit the optical view through transparent covering materials in addition to simpler fluidic control, parallelization, and multiplexing. 

Recently, we adapted CLEF to these substrates by conceiving a wireless electrochemiluminescent (ECL) planar micropore in a microfluidic device (Figure 10) [116]. The microfluidic conception combined with selective etching of the silicon oxide in the micropore region permitted using two orders of magnitude lower voltages for generating ECL signals from the silicon micropore compared to standard bipolar electrochemistry setups. The planar pore approach combining CLEF and microfluidics at the level of solid-state micropores is very promising. The ease of making a series of planar pores, possible combination with other lab-on-a-chip functions such as sample pretreatment, and/or parallelization of functionalized pore sensors open the way to new kinds of biosensing platforms for multiplexing.

## 3. Conclusions and Prospects

Contactless electro-functionalization is an innovative methodology to achieve the localized grafting of various electroactive entities exclusively on the inside walls of micro- and nanopores. This very versatile technique overcomes the challenge of selectively functionalizing the inside walls of a pore manufactured in a solid-state dielectric membrane. Numerical simulations provided relevant cartographies of the electrical field in the pore environment. The developed numerical model was validated by achieving a good correlation between simulated and experimental impedance spectra. Two necessary conditions for CLEF efficiency are the presence of a sandwich membrane, made of silicon covered with silicon oxide, and a threshold value for the applied voltage. Above this threshold value, the sandwich-like structure of the membrane induces the polarization of the dielectric silicon oxide, and the applied voltage creates a pathway for charge carriers between the inner silicon core and the electrolyte. The major role of the silicon inner layer was also corroborated by the major influence of light on the CLEF process. CLEF finds promising applications in biosensing, particularly in living cell analysis. The presence and the identification of B- and T-lymphocytes was achieved using a micropore functionalized via CLEF with specific antibodies whereas these cells are indistinguishable by optical microscopy without prior labeling. This opens opportunities for the analysis of other types of cells. We anticipate that CLEF will open the way, in the next years, to new electrochemically inspired methodologies for localized grafting in micro- and nanopores.

The limitations of the CLEF methodology lie mainly in the relatively complex experimental setup needed for its implementation. A solution to simplify the experimental process is to switch to “planar” pores, far easier to conceive and parallelize than membrane-through pores and to which CLEF could be adapted.

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
