# Peer review of "Polarization Induced Electro-Functionalization of Pore Walls: A Contactless Technology"

_biosensors, 2019, doi:10.3390/bios9040121_

Round 1

Reviewer 1 Report

Bouchet-Spinelli and el al, provide recent development of the sensing by nanopore devices containing both bio and solid-state pores. In addition, the authors focus on a functionalization technique of a nanopore, called CLEF, which allows to deposit various materials only inside of the nanopore wall. Although the manuscript is generally well written and clearly presented, there are numerous comments below, which the authors would consider in order to publish it. Mainly the choice of references seems quite questionable. If the authors intended to provide recent progress of the nanopore technology, the authors would mention the difference between the original works, cited most by nanopore articles and references chosen by the authors. Furthermore, the writing style of the chapter 3 is likely a research article, which has to be reviewed as a “research article.” In brief, I would recommend the chapter 3 as a research article although the manuscript has to be resubmit as a new manuscript. If the authors publish as a review article, the chapter 3 would move into chapter 2 as 2.3 or 2.4 and revise the manuscript with proper references. Major comments: The manuscript has been submitted to publish as “review.” Indeed, 1. Introduction and 2. Principle of.. containing 2.1, 2.2, 2.3 describe a basic overview of nanopore technology with the recent development. However, especially in 2.2 solid state pores, the references are not well selected. If this manuscript is review, the authors describe the historical nanopore articles and the development first and then recent progress. For example, the authors mention graphene nanopore with the representaive reference by Qi. The article is specifically interesting in the selective ion kinds through graphene pores. On the other hand, everyone working on solid state nanopore knows Jene Golovchenko (~2018) published an article on Nature. I remember another article was published nearly the same day by a group at Penn. Here, the other examples, In line 116: About “focused ion beam[59],” I would not describe it with these words. Jene’s first nanopore is not made by FIB but Ar beam although it was focused. The [59] is the most cited article related to the solid state nanopore devices. In the manuscript on 2.2, [54] or [55] seems to be the historical articles for the solid state nanopore. Are they review articles? There are a few major review articles of nanopore along with [1], [16]. For example, Wanunu’s article is well written the nanopore history. I recommend the authors look through the article. As the authors may know that the researchers working on nanopores are very strict to choose the reference. About 3 the CLEF technology, the number of reference is only 3 or 4 even the chapter 3 extends over 4 pages. The manuscript seems to be a research article providing preliminary results by the authors. Minor comments: In line 18, It starts with “we have developed” in the Abstract. The sentence would not fit to a review article. In line 37, it goes as A, B, C, D …..[1-15]. To be a review article, A[], B[], C[] and D[] is better. Is the typing “...“ OK as a scientific writing style? In line 55, should the [16] and [17] be replaced? I do not follow the sentence. [16] is a major review article, which may appear in Intro. In line 94, “atomistically defined structure “ would be just atomically defined structure of atomically well-defined structure. Below line 115, the authors may reconsider the references again. Some references are not original but new or the authors should mention the reason to choose them. I would describe the nanopore history first and then explain related recent progress with the details. In line 120, Al2O3[59] is not correct. In 2.3, The authors mentions Yusko et al, Nature Nanotechnol, 6, 253, 2011 although the other Yuslo’s article is refereed. Bellow the chapter 3, the manuscript should be revised as a research article as this reviewer mentioned but as comments, In line 196, the authors mention the CLEF technology can be applied to the pore with the diameter from micro to nano meter scales supported by numerical simulations using COMSOL. The aspect ratio between the scale is far different induce access resistance at pores so numerical simulation should be performed both length scales. In line 220, how is 1 MHz incremented into the numerical simulation? Is the COMSOL execute stationary? In line 221, this reviewer does not follow the idea to present the Er component only to discuss the electric field. In the potential plots in Fig 2 B, C indicate Er is very small compare to the Ez. Bumpy nanopore inner wall would produce such Er’s but the ions inside pore are affected by E field as a vector not the single component of r. Presumably the E field near pore inner wall is in a direction perpendicular to the surface. In line 234, “ the Complex Electro kinetic equation “ must be delated. The equation would be Posson and Nernst-Planck equations. This reviewer think the Naview stokes is not included here. The author should describe it. I recommend the authors provide each parameter as SI, if the chapter 3 is resubmitted as research paper. In line 246, this reviewer recommends deleting “the polarization~”. I do not see any difference between B and C outside of the pore membrane, meaning no influence for the CLEF technology. Between B and C, I see no difference with our without calculating In 3.2.2, the authors discuss the impedance measurements. The authors should provide resistivity of the Si wafer with the doping kind. Is the Si ground?

Reviewer 2 Report

Authors present the very promising Contactless Electro-Functionalization (CLEF) method for pore wall functionalization and subsequent sensing of single biomolecules and cells in the micro- and nano-sized pores. The complex membrane structure (Si/SiO2) becomes a key in the electrochemical drafting onto the pore walls, and incorporation of the electric contacts into the solid-state pore design further helps sensing. The selective functionalization can be with the different types of structures such as gold nanoparticles, oligonucleotide-functionalized polypyrrole and further addition of antibody conjugates to PPy-ODNs that open way to the sensing of a variety of molecule and cell types as it was demonstrated in the experiments. The functionalization was confirmed, and the mechanism was explained by the polarization of the SiO2 membrane as it was shown in numerical simulation. The prospective functionalization technology can be applied to different types of pores in micro- and nanofluidic devices and can serve in a number of the application.

Several issues that would be beneficial to discuss for technology development such as

-        Dependence of the electric field repartition vs. the SiO2 thickness, which thickness may be especially important in pores of nanoscale diameter,

-        Electric field amplitude threshold for different pore sizes remains the same at functionalization or depends on parameters of the pore, (in relation to Fig.4)

-        The density of deposited functional structures (nanoparticles or PPy-ODNs) in micro- and nanopores to predict possible sensing events of the complimentary targets relative to the general flow through the functionalized pore at the translocation time (in relation to Figs.7 and 8).

Round 2

Reviewer 1 Report

The revised manuscript is fair to publish as a review article. The choice of the references seems reasonable. However, this reviewer still 2.4.2 is not well reviewed. If 2.4.2 has been published elsewhere, site the article. If not, maybe it is OK to present the author’s brand-new results in such a review article in order to explain some mechanism. The author’s reply to the reviewer’s comments is fairly reasonable around the 2.4.2 so why not the authors revise the manuscript based on the reply. The other sections are fine.

About my previous comment, “I do not see any difference between B and C outside of the pore membrane, meaning no ~~” is about figure 2. Now I see electric fields at vicinity of the pore as a vector. It seems that the locations of vectors have been changed from previous manuscript. And yes, it looks as the authors claim. However, as for inside the membrane, E field inside the membrane is calculated only for C, not for B. It is likely no triangle meshes inside the membrane for the FEM calculation for B, resulting in no color for Er. Er = 0 would be green. Instead, the authors mention that “if the pore membrane is considered as a perfect insulator, the electrical field is entirely concentrated inside the pore (Figure 2B).” This sentence sounds scientifically odd. If the authors meant there is no electric field inside a membrane material, the material could be a superconductor. It could be just conductive materials, in which potential is uniform. If the membrane is an insulator, simply the authors must calculate as an insulator by comsol. Any insulator material produces E field inside because it is insulator. Then, the authors claim an electrical connection between electrolyte and silicon substrate inside the SiO2 layers by applying an AC bias. I would think it is not necessary to drive current via SiO2 because of the AC bias and the capacitive effect described in 3.2.2. In addition, how high the “high electrical voltage” is in the line 255. "However, CLEF does not work under this experimental configuration" in line 244 means the exp. config. like figure B. Is there any reference for this claim?  

Finally, this reviewer stops mumbling about the section and again recommend this manuscript for publication as a review article after all the anthers check the section.  

Besides, this reviewer recommends such conductive Si substrate to be grounded for this CLEF application.

Round 3

Reviewer 1 Report

The revised manuscript is now looking sharper. As for 2.4.2 with Fig. 2, I am very satisfied with the authors new figures and their descriptions concerning the electric field inside the pore of Fig. 2.